# What Is Parvalbumin for?

**DOI:** 10.3390/biom12050656

**Published:** 2022-04-30

**Authors:** Eugene A. Permyakov, Vladimir N. Uversky

**Affiliations:** 1Institute for Biological Instrumentation of the Russian Academy of Sciences, Federal Research Center ‘Pushchino Scientific Center for Biological Research of the Russian Academy of Sciences’, Pushchino 142290, Moscow Region, Russia; 2Department of Molecular Medicine, Morsani College of Medicine, University of South Florida, Tampa, FL 33647, USA

**Keywords:** parvalbumin, oncomodulin, structure, stability, calcium binding, physiological functions

## Abstract

Parvalbumin (PA) is a small, acidic, mostly cytosolic Ca^2+^-binding protein of the EF-hand superfamily. Structural and physical properties of PA are well studied but recently two highly conserved structural motifs consisting of three amino acids each (clusters I and II), which contribute to the hydrophobic core of the EF-hand domains, have been revealed. Despite several decades of studies, physiological functions of PA are still poorly known. Since no target proteins have been revealed for PA so far, it is believed that PA acts as a slow calcium buffer. Numerous experiments on various muscle systems have shown that PA accelerates the relaxation of fast skeletal muscles. It has been found that oxidation of PA by reactive oxygen species (ROS) is conformation-dependent and one more physiological function of PA in fast muscles could be a protection of these cells from ROS. PA is thought to regulate calcium-dependent metabolic and electric processes within the population of gamma-aminobutyric acid (GABA) neurons. Genetic elimination of PA results in changes in GABAergic synaptic transmission. Mammalian oncomodulin (OM), the β isoform of PA, is expressed mostly in cochlear outer hair cells and in vestibular hair cells. OM knockout mice lose their hearing after 3–4 months. It was suggested that, in sensory cells, OM maintains auditory function, most likely affecting outer hair cells’ motility mechanisms.

## 1. Introduction

Parvalbumin (PA) is a small, acidic, mostly cytosolic Ca^2+^-binding protein of the EF-hand superfamily. It was found in lower and higher vertebrates, including humans (for reviews, see [1,2,3,4]). PA was found in fast twitch muscle cells, heart tissue, nephrons, specific neurons of the central and peripheral nervous system, certain cells of several endocrine glands, and sensory cells of the mammalian auditory organ, the organ of Corti, and some other cells.

Intracellular PA was suggested and partially proved to serve as a soluble relaxing factor accelerating the Ca^2+^-mediated relaxation phase in fast muscles (reviewed in [1]). It is assumed that PA serves as a slow metal buffer in non-muscle tissues, modulating flows of metal ions. An extracellular form of PA was found in some biological systems [5,6,7]. Biological functions proposed for extracellular PA include antibacterial, chemoattractant, and immunomodulatory actions.

Despite many years of research on PAs, which have provided a good understanding of their structure and physicochemical properties, the physiological functions of PA are still far from being clear. This short review covers some new findings on structural and physicochemical properties of PAs and focuses on their potential physiological significance.

## 2. Parvalbumin Structure

PA consists of 106–113 amino acid residues and its molecular mass is ~11–12 kDa. Isoelectric points of PAs are within the range of 3.9 to 6.6 (see Table 1). PAs are characterized by very low content of Trp (a single residue only in several fish species) and Tyr (absent or present as 1 or 2 residues), but high content of Phe residues (9–11 per molecule). These amino acid composition biases are reflected in the characteristic ultraviolet absorption spectra of PAs with clearly visible fine vibrational peaks of the Phe chromophores.

The PA family diverged into α and β sub-lineages [8,9,10]. β-PAs are also referred to as ‘oncomodulins’. The α and β isoforms differ in isoelectric point (α: pI > 5; β: pI < 5) and show differences in amino acid residues in at least 11 positions [10]. β-PAs are characterized by the presence of Cys in the loop of the AB domain and the absence of an additional C-terminal residue in the C-terminal helix. Despite the fact that the primary structures of α and β PAs differ significantly, their three-dimensional structures are very close. 

PA molecule is a compact globule composed mostly of α-helices and a small amount of β-strands (reviewed in [1]) (Figure 1). Kretsinger and coworkers [11,12] determined the crystal structure of carp β-PA and revealed that a Ca^2+^-binding domain is composed of two α-helices linked by a Ca^2+^-binding loop creating a structural motif, which is currently known as ‘EF-hand’.

Ca^2+^-binding loop of the EF-hand usually contains the most commonly observed consensus sequence DxDxDG. The negatively charged Ca^2+^-liganding oxygen atoms donated by carboxyl and carbonyl groups are located at vertices of a distorted octahedron. The side chains of loop residues at positions 1, 3, 5, and 12 provide the +*x*, +*y*, +*z*, and −*z* oxygen ligands, respectively. Water molecule oxygen is located at the −*x* position, and a backbone carbonyl oxygen occupies the −*y* position. In some cases, the carboxylate at −z position coordinates calcium by both oxygens and the octahedral geometry of the Ca^2+^ binding site turns into a distorted pentagonal pyramid geometry.

PA molecule contains two active EF-hands, CD and EF domains. The AB domain of PA cannot bind Ca^2+^, since its AB-loop is too short and the DxDxDG motif is disturbed. Meanwhile, the AB domain shields the hydrophobic core of the protein and the hydrophobic parts of the functional EF-hands from solvent and influences their calcium affinities [13,14,15]. The CD and EF loops are connected by a short antiparallel β-sheet between them. Conservative Arg75 and Glu81 residues (numbering according to carp β-PA) form a salt bridge protected from solvent by the AB domain (see Figure 2).

Some of the PAs contain cysteine residues (up to four, as in β-PA of Atlantic salmon), but they do not form S–S links under the non-oxidizing conditions.

Comparative analysis of the spatial structures of proteins of the EF-hand superfamily, including PA, made it possible to discover two highly conserved structural motifs, which form a supporting scaffold for the Ca^2+^-binding loops [16]. Each conserved structural motif forms a cluster consisting of three amino acids: cluster I (‘black’ cluster) and cluster II (‘grey’ cluster) (Figure 2 and Figure 3). Cluster I (‘black’) is most conserved and consists of mostly aromatic amino acids. The cluster is stabilized by a set of linked CH-π and CH-O hydrogen bonds between the side chains of the amino acids. Cluster I was suggested to be important for structural stabilization of the paired EF-hand domain in its region, where the polypeptide chain enters and exits the domain [16]. In contrast, cluster II consists of a mix of aromatic, hydrophobic, and polar amino acids. It is much less conserved and lacks stabilizing interactions. It has been suggested that the higher variability of cluster II (‘gray’) could facilitate adaptation of the EF-hand domain to various conformational and dynamic changes [16].

Figure 3 shows three-dimensional (3D) structure of rat β-PA (oncomodulin) with its clusters I (‘black’) and II (‘gray’), consisting of F48, A100, and F103; and G61, L64, and M87, respectively. The amino acids of the clusters I and II were sequentially substituted by alanine and physical properties of the resulting mutant proteins were studied [17]. In spite of rather complicated patterns of effects of separate residue substitutions in the clusters I and II, the Ala substitutions in cluster I cause noticeably more pronounced changes in various structural parameters of proteins, such as hydrodynamic radius of apo-form, thermal stability of Ca^2+^- and Mg^2+^-loaded forms, and total energy of Ca^2+^ binding in comparison with the changes caused by similar amino acid substitutions in the cluster II. In agreement with these experimental results, it was found that local intrinsic disorder propensities and the overall levels of predicted disorder in rat β-PA are strongly affected by mutations in cluster I, whereas mutations in cluster II generate less pronounced effects [2]. These results demonstrated that the amino acids of the cluster I provide more essential contribution to the maintenance of structural and functional properties of the protein in comparison with the residues of the cluster II.

## 3. Binding of Metal Ions to Parvalbumin

The general Figure 1 describing the binding of Ca^2+^ by a protein with two binding sites is as follows [1,18]:

In the Figure 1, P is a protein. It means that, in general cases, the binding of Ca^2+^ to PA occurs in two ways when the CD and EF binding sites are filled by Ca^2+^ simultaneously. At the same time, experiments carried out on mutant human and rat parvalbimins demonstrate that the substitution of one of the Ca^2+^ chelating Glu by Gln in the CD site completely abolishes the high metal affinity of parvalbumins, whereas similar mutation in the EF site abolishes the high affinity metal binding to the EF site and only causes a decrease in the metal affinity of the CD site by an order of magnitude [19]. This suggests that the binding of Ca^2+^/Mg^2+^ to parvalbumin is a sequential process (1) and the CD site is occupied first
P ↔ PCa ↔ CaPCa(1)

The results of many other studies carried out by various methods also suggested the sequential mechanism of Ca^2+^ binding to PA (reviewed in [1]). At the same time, there is a disagreement in the literature regarding the sequence of the filling the CD and EF binding sites with calcium. According to our experimental data, the CD site seems to be filled by Ca^2+^ first, followed by the filling of the EF site [19]. Pauls et al. [20] found that inactivation of the EF site in rat PA, much more than of the CD site, impairs divalent cation binding. These discrepancies can be explained by the fact that Pauls et al. studied mutants with multiple substitutions, which were very different from those used in our work and under very different conditions.

PAs are characterized by extremely high affinity to Ca^2+^ ions. In fact, equilibrium Ca^2+^-association constants of PAs measured at low ionic strength lie in the range from 10^8^ to 10^10^ M^−1^ (reviewed in [1]). Mg^2+^ ions compete with Ca^2=^ ions for the same two binding sites. The equilibrium Mg^2+^-binding constants of PAs measured at low ionic strength are within the range from 10^4^ to 10^5^ M^−1^ (reviewed in [1]). PA also binds Na^+^ and K^+^ ions with equilibrium association constants in the range from 0.1 M^−1^ to 50 M^−1^, and Na^+^ binds an order of magnitude more strongly than K^+^ (reviewed in [1]).

Equilibrium metal association constant K_a_ is the ratio of the kinetic constants of association (k_on_, M^−1^s^−1^) and dissociation (k_off_, s^−1^) (2):K_a_ = k_on_/k_off_(2)

Fluorescent stopped flow method was used to study kinetics of Ca^2+^ and Mg^2+^ dissociation from cod PA after rapid mixing of the metal bound protein with a strong metal chelator, EDTA [21]. Analysis of the experimental kinetic curves showed that the k_off_ values for Ca^2+^ and Mg^2+^ are rather close to each other in the temperature region from 10 to 30 °C. Two exponential terms in the kinetic curves were interpreted as Ca^2+^/Mg^2+^ dissociation from the two binding sites of PA. The values of dissociation rate constants obtained from these analyses at temperatures from 10 °C to 30 °C are within the range 0.03–0.8 s^−1^ and 0.2–5 s^−1^ for Ca^2+^-PA complex and 0.9–4.5 s^−1^ and 4–33 s^−1^ for Mg^2+^-PA complex. It means that the differences in the equilibrium binding constant values for Ca^2+^ and Mg^2+^ (K_a_) are mostly due to the differences in the k_on_ values. The k_on_ values for Ca^2+^ calculated from these data are within the range from 10^6^ to 10^9^ M^−1^s^−1^ (temperature range 10 to 30 °C); i.e., they are close to the diffusion-controlled limit. The k_on_ values for Mg^2+^ are 2 to 5 orders of magnitude lower. The drastically lower k_on_ values for Mg^2+^ this can be explained by the fact that Mg^2+^ ion in aqueous solution is much more strongly hydrated compared to Ca^2+^ ion due to the fact that it has 3.5 times higher charge density.

Analysis of the 3D structures of the EF-hand proteins with and without bound calcium showed that the binding of calcium does not dramatically change interactions within cluster I (contact area with and without bound calcium averaging 89 ± 12 Å^2^ and 84 ± 11 Å^2^, respectively) [16]. In some EF-hand Ca^2+^ binding proteins, the interaction with calcium also slightly changes the contact surface area in the cluster II (68 ± 12 Å^2^). In the other group of Ca^2+^ binding proteins, calcium binding results in a dramatic increase in the interactions between the three residues of cluster II (contact surface area 11 ± 8 Å^2^). PA belongs to the first group of proteins: the binding of Ca^2+^ does little to change the interactions in both conservative clusters [16]. 

Removal of bound metal ions from PA (formation of apo-form) makes its structure looser and makes many of its groups more accessible to the solvent (reviewed in [1]). The removal of Ca^2+^ from PA results in a decrease in its α-helical content [22,23,24]. Interestingly, character of the Ca^2+^-induced changes in α-PAs and β-PAs is different [25,26]. Rat α-PA in the apo- and Ca^2+^-loaded state has approximately the same three-dimensional structure. Ca^2+^ binding to α-PA causes changes mainly in the structure of the loop regions. In contrast to this, Ca^2+^ binding to rat β-PA results in a conformational change in the AB domain; rotation of the C, D, and E helices; changes in interdomain contacts; and rearrangement of the hydrophobic core. However, all these changes do not affect the overall protein folding. In the Ca^2+^-loaded rat β-PA, Phe70 is buried in the protein core, but in the apo-protein is located near the protein surface. Similarly, the single Trp of cod or whiting β-PA is buried deep inside the Ca^2+^-loaded protein, but Ca^2+^ removal causes a movement of this Trp to the protein surface [27]. 

Thermal stability of apo-PA is low: mid-transition temperature, *T_m_*, of apo-PAs under low ionic strength is within the range from 30 °C to 50 °C [28,29,30]. Calcium binding causes a pronounced shift of thermal transition to higher temperatures (up to 120 °C in some cases) [27]. Interestingly, the scanning calorimetry method revealed the lack of fixed tertiary structure in some apo-PAs, for example in pike apo-α-PA, which is reflected in the absence of ‘all-or-none’ thermal transitions for these proteins [24]. This is consistent with the facts that Ca^2+^ removal from such PAs decreases their α-helical content, and increases mobility of the aromatic residue environment, monitored by near-UV circular dichroism method. These results show that pike apo-α-PA is in the classical molten globule state [31,32,33]. At the same time, the apo-form of pike β-PA is characterized by a much less conserved secondary structure, and this does not allow us to attribute this state to the classical state of molten globule. Intrinsic disorder analysis of PAs with folded and unfolded apo-state using the predictor of naturally disordered regions PONDR^®^ VSL2 showed that the N-terminal region of PA—including α-helix A, AB-loop, and N-terminal half of α-helix B—was predicted to be less ordered in PAs with disordered apo-state [34]. It has been shown that PAs with disordered apo-state comprise about 16–19% of all PAs.

Thermal denaturation of Ca^2+^- and Mg^2+^-loaded states of five PAs (cod PA, α and β isoforms of pike, and rat PAs) was studied by scanning calorimetry method in a temperature range from 15 °C to 125 °C [29]. The proteins were divided into three groups according to their different thermal behavior. The proteins of the first group have a single heat sorption peak in their Ca^2+^-loaded state (rat PAs). The second group is characterized by two distinct heat-sorption peaks in their Ca^2+^-loaded state (cod PA and pike α-PA). The unfolding behavior of the proteins of the last group (pike β-PA) above 100 °C is complicated by an oligomerization process. An analysis of the thermal denaturation curves of the Ca^2+^-bound PAs of the first group showed that their unfolding is not described by the simple ‘all-or-none’ scheme, and the process proceeds with a formation of an intermediate state. The multistage character of the thermal unfolding is evident in the melting curves of the PAs of the second group (pike α-PA and cod PA). They are characterized by two distinct heat-sorption peaks separated by 20–30 °C [29].

In contrast to Ca^2+^-bound state, Mg^2+^- and Na^+^-bound states of pike α-PA are characterized by a single heat sorption peak [29]. Mg^2+^ binding causes expectedly less pronounced stabilizing effect in comparison with that induced by Ca^2+^ binding (*T_m_* is *ca* 77 °C in 1 mM MgCl_2_, compared with 90 °C in 1 mM CaCl_2_). Na^+^ binding results in even less pronounced stabilization (*T_m_* is *ca* 33 °C in 300 mM NaCl). The thermal unfolding of both Mg^2+^- and Na^+^-loaded states of pike α-PA obeys the simple ‘all-or none’ mechanism. 

These results show that the exchange of Mg^2+^ by Ca^2+^ ions in the PA binding sites seriously alters its thermodynamic properties: the simple ‘all-or-none’ transition converts into a more complex multistage transition. Moreover, depending on its interactions with metal ions, PA (for example, pike PA) can be an intrinsically disordered protein (apo-form) or an ordered mesophilic (Na^+^-bound state), thermophilic (Mg^2+^-bound state), or even hyperthermophilic (Ca^2+^-bound state) protein [29].

The additional peak in the heat-sorption curves of some Ca^2+^-loaded PAs seems to arise due to stabilization of an intermediate state, which is absent in PAs exhibiting single peak in their heat-sorption curves. It has been found [29] that the structure of pike α-PA, which demonstrates two heat sorption peaks during thermal unfolding, is notably different from the structure of three other PAs studied: it has nine interior cavities located mostly between the EF and CD subdomains versus four to six in other PAs. Moreover, the total area (S) and volume (V) of the cavities in pike α-PA exceeds the S and V values of other PAs by more than 40%. This indicates that the overall packing density of pike α-PA is essentially lower than that of other PAs. The PONDR^®^ analysis showed the EF subdomain is included in the most stable thermodynamic domain in pike α-PA, whereas the CD and AB subdomains are contained in the less stable thermodynamic domain.

It is well known that Ca^2+^/Mg^2+^ exchange in the binding sites of PAs results in alternation in the metal ion coordination mode: the coordination number decreases from 7 in the Ca^2+^-bound state to 6 in the Mg^2+^-bound state [35,36,37]. The Glu residue at position 12 in the EF-hand loop sequence (gateway ‘Glu12′) acts as a bidentate ligand in the Ca^2+^-bound state and as a monodentate ligand in the Mg^2+^-bond state, which can change the protein conformation and seriously affect thermodynamic properties of the protein [29]. 

## 4. Interactions with Peptides and Membranes

So far, no target proteins have been found for rat or mouse PA. At the same time, it was shown that under certain conditions, PA is able to interact with short peptides. It was found that pike PAs pI 4.2 and 5.0 bind amphiphilic bee venom peptide melittin (26 amino acid residues) [38]. In apo-state, the PAs form a tight equimolar complex with melittin (binding constant 10^6^ M^−1^ at 18 °C), while their Ca^2+^- and Mg^2+^-loaded states do not bind this peptide [38]. Heating of the apo-PAs up to the temperatures above their thermal unfolding transition does not change the stoichiometry of the complexes, but increases their association constants by an order of magnitude (binding constant 10^7^ M^−1^ at 44 °C). Isolated Ca^2+^-binding fragment 38–108 of pike PA pI 5.0 (CD-EF) retains the ability for Ca^2+^-inhibited equimolar binding of melittin. 

In contrast, the well-known Ca^2+^-sensor protein of the EF-hand family, calmodulin, binds melittin with a very high binding constant 10^10^ M^−1^ in the Ca^2+^loaded state [39,40]. In the absence of Ca^2+^, calmodulin binds melittin with a much lower affinity, with the binding constant 10^5^ M^−1^ [40].

Since both intracellular and extracellular PA is always found in an environment with high concentrations of divalent calcium and magnesium cations, it almost never exists in the apo-state. For this reason, its interaction with the amphiphilic peptide, discovered in [38], is unlikely to have a physiological significance.

It has been found that cod and pike PAs interact with model synthetic (dipalmitoylphosphatidylcholine, DPPC) and natural (phosphatidylcholine and phospatidylethanolamine) vesicles (mean diameter 300–500 Å) [41]. It has been found that the binding of Ca^2+^ and Mg^2+^ ions to PA modulates its interaction with the DPPC vesicles. The interaction of the liposomes with Ca^2+^-bound PA shifts its thermal transition towards higher temperatures by 2 °C to 3 °C, while the interaction of the liposomes with the metal-free and Mg^2+^-bound protein results in similar shift of the thermal transition but towards lower temperatures. Moreover, the interaction of the protein with the liposomes causes changes in the calorimetric heat sorption peak of model synthetic liposomes. The denaturation curves measured by tryptophan intrinsic fluorescence method for free and liposome-bound cod PA are strikingly different. The thermally induced red spectral shift for the liposome-bound protein, which reflects an unfolding of the protein, begins at 10 °C lower temperatures than that for free PA, but the shift magnitude is less pronounced in this case. Tryptophan residue in the thermally denatured liposome-bound cod PA remains inaccessible to water, while the thermal unfolding of the free protein causes a translocation of its single Trp to the protein surface.

Interestingly, PA immunoreactivity was found in (or near) membranous systems, such as mitochondria and (or) microtubule in male gonads of *Drosophila melanogaster* [42].

## 5. Possible Functions of Parvalbumin

### 5.1. Parvalbumin in Muscle Cells

Of the possible biological functions of PA, perhaps the one that has been explored the most is that in muscle cells. It has long been known that PA concentration in muscles correlates with the speed of muscle contraction and relaxation. PA concentration in sarcoplasm of fast skeletal muscles reaches a millimolar level, while slow-twitch muscles are characterized by about two orders of magnitude lower levels of PA [43,44,45]. The positive correlation between PA content and relaxation rate of muscles was found and this led to the assumption that PA could facilitate Ca^2+^ translocation within the sarcoplasm and therefore could be a relaxation factor in fast-twitch muscles [43,44].

Free Ca^2+^ concentration in sarcoplasm of a resting muscle is very low (<1 μM), while free Mg^2+^ concentration in the muscle is in the millimolar region. For this reason, PA in the sarcoplasm of a resting muscle seems to be loaded by Mg^2+^ ions. Ca^2+^ transiently released from the sarcoplasmic reticulum diffuses through the sarcoplasm and activates muscle contraction interacting with troponin complex located on the actin fibrils. PA is located in the sarcoplasm of muscle cells right on the way of the Ca^2+^ movement, but since the exchange of Mg^2+^ for Ca^2+^ in the binding sites of PA is governed by the relatively slow process of Mg^2+^ dissociation, Ca^2+^ ions can reach the Ca^2+^-specific sites of troponin C [46]. Later on, the strong Ca^2+^/Mg^2+^-sites of PA will compete for Ca^2+^ and promote its removal from troponin C, which will start muscle relaxation. Since the dissociation rate of Mg^2+^ in PA is low, it is reasonable to think that PA will not have time to remove calcium during a single contraction–relaxation cycle (twitch), especially in fast muscles. Instead, the Mg^2+^-Ca^2+^ exchange in PA can only happen after many successive twitches (after tetanus). Below are the results of some experiments indicating the important role of PA in accelerating the relaxation of fast muscles.

Cross-reinnervation of slow- and fast-twitch muscles results in a decrease in PA content in the fast-twitch muscle, and to an increase in PA content in the slow muscle [47,48]. PA concentrations in fast muscle fibers of small mammals are higher than in fast muscle fibers of big mammals because of the essential difference in their contraction and relaxation rates [49].

Direct gene transfer experiments were carried out to check whether PA really could work as a relaxing factor. PA cDNA was transferred in vivo to normal and regenerating rat soleus muscles, which do not normally synthesize PA [50]. Considerable concentrations of PA mRNA and PA appeared in uninjured and regenerating muscles two weeks after the transfection, which caused a shortening of half-relaxation time of these muscles. In another work, it was found that knock out of PA gene in mice fast-contracting/relaxing muscles results in slowing down the decay of Ca^2+^ level after a 20-ms stimulation of the isolated exterior digitorum longus (33% lower rate constant of Ca^2+^ decay), and in a reduction in the time to reach peak twitch tension [51].

It was found that the relaxation rate of extensor digitorum longus in PA-deficient mice at 20 °C is low and does not depend on tetanus duration (<3.2 s) [52]. In contrast, the relaxation rate of normal wild type muscles decreases when tetanus duration increases from 0.2 to 3.2 s and the fast relaxation is restored with increasing rest interval. The slowing of the relaxation stage caused by the tetanus duration increase was explained by a saturation of PA by Ca^2+^, while the fast relaxation recovery after the increase in rest interval by a formation of Ca^2+^-free PA. In flexor digitorum brevis muscles, the effects of tetanus duration on wild type and PA-deficient muscles were qualitatively similar to those observed in extensor digitorum longus muscles. The authors concluded that PA accelerates calcium movement from myofibrils to sarcoplasmic reticulum, mostly after tetanus. 

The superfast toadfish swimbladder muscle, the fastest vertebrate muscle known [53], is characterized by the highest PA concentration (1.20–1.35 mM) [54,55]. Toadfish produce sounds by their swimbladder muscles which contract at frequencies exceeding 100 Hz. Toadfish produce a 400 ms high frequency call followed by a 5–15 s intercall interval [54]. It has been proposed that PA rapidly binds most of the Ca^2+^ released during the call and this Ca^2+^ moves back into the sarcoplasmic reticulum during the long intercall interval [55]. Midshipman (*Porichthys notatus*) is another fish which produces sounds with a frequency 80–100 Hz at a temperature of 12–15 °C with a 100% duty cycle without any intercall intervals. In this case, PA seems to be of little use as a relaxation factor since it would become saturated at an early stage of calling. Tikunov and Rome [55] found that the midshipman swimbladder muscle still has high PA concentration (0.18 mM). Interestingly, total PA content in calling male midshipman swimbladder and PA content in swimbladder of non-calling female and much slower locomotory muscles are practically identical. These results show that PA does not play the role of relaxation factor in the swimbladder muscle of midshipman fish. Moreover, these observations raise some doubts that the main function of parvalbumin in fast muscles is to accelerate their relaxation.

In addition to fast skeletal muscles, PA has also been found in cardiac muscle tissue of rat, mouse, chicken, rabbit, and pig [56,57], and it was revealed that PA takes part in the relaxation of cardiac myocytes [58,59,60,61,62,63]. It has been suggested [60] that at the end of systole, Ca^2+^ dissociates from troponin C and binds to PA, which causes cardiac myocyte relaxation. Now PA is tested as a pharmacological agent to treat heart dysfunctions. For example, it has been found that PA fully normalizes the relaxation rate in diseased cardiac myocytes taken from an animal model of human diastolic dysfunction [58]. PA gene transfer to the heart in vivo results in PA expression up to the levels similar to those in fast skeletal muscles. The synthesized PA accelerates heart relaxation in normal hearts and corrects heart relaxation in an animal model of slowed cardiac muscle relaxation [59]. Similar results were obtained by Michele et al. [61]: they found that PA gene transfer and expression in vivo increased relaxation rate in the aged myocardium. Asp et al. [64] used mutant PA E101D (monodentate instead of bidentate Ca^2+^ coordination in the CD binding site) to treat diastolic heart failure. E101D PA is characterized by 114-fold decreased Ca^2+^ affinity and 28-fold increased Mg^2+^ affinity compared to the wild type PA. E101D PA increased contraction amplitude of myocytes compared to both untreated myocytes and myocytes with E101Q PA, with slight improvements in relaxation. Moreover, E101D PA increased spontaneous contractions after pacing stress.

Summarizing, we can conclude that a lot of experimental data indicate that in many cases, PA can accelerate the relaxation stage of fast muscles, but this effect is not too pronounced, as the PA gene knock out results in 33% lower rate constant of Ca^2+^ decay [51]. 

Interestingly, fast muscles with high PA concentrations are characterized by the highly efficient oxygen consumption (oxygen uptake up to 57 mL/min [65]). The molecular oxygen is utilized for oxidative phosphorylation in mitochondria for production of ATP molecules. During the consumption by mitochondria, up to 0.5% of oxygen escape from the electron-transfer chain (complexes I and III) and is reduced with the formation of superoxide anion (O_2_^●–^). Later O_2_^●–^ is converted to hydrogen peroxide (H_2_O_2_) (reviewed in [66]). H_2_O_2_ is one of the intracellular reactive oxygen species (ROS). It is most abundant (0.1 μM) and long living (half-life time of 10 μs) ROS, which acts both as an inducer of oxidative damage and a signaling molecule [65]. The major intracellular source of ROS is mitochondrial respiration. The PA-rich cells are usually characterized by high oxygen consumption and intense production of ATP and ROS, including O_2_^●–^ and H_2_O_2_. It is of interest that PA immunoreactivity was found in (or near) mitochondria [42].

We have explored the possibility that PA could serve as a free radical scavenger. ORAC (oxygen radical absorbance capacity), TEAC (trolox equivalent antioxidant capacity), and hydrogen peroxide AOC assays were used to study antioxidant capacities (AOC) of various forms of intact rat α-PA [67]. We have found that the oxidation of PA depends on its conformation: AOC value for apo-PA (similar to AOC for the proteolized protein) 4–11-fold exceeds AOC for the Ca^2+^-loaded protein, while AOC for Mg^2+^-bound PA is similar to that of apo-PA. The interactions of PA with ROS causes oxidation of its Phe residues. It has been found that total antioxidant capacity of PA under in vivo conditions may reach the level of reduced glutathione. For this reason, one can suggest that one more physiological function of PA in fast muscles could be a protection of these cells from reactive oxygen species. Moreover, PA could change intracellular redox equilibria and signaling in a Ca^2+^-dependent manner.

### 5.2. Parvalbumin in Neurons

Calcium binding proteins of the EF-hand superfamily, parvalbumin, calbindin, and calretinin, have been found in various classes of inhibitory interneurons as well as in some pyramidal neurons in the mammalian neocortex (see [68,69] for reviews). Their physiological role is assumed to be Ca^2+^ buffering, Ca^2+^ transport, regulation of activity of various enzymes, and a protection of neurons against calcium overload. It is assumed that neurons containing high concentrations of these Ca^2+^-binding proteins would be more resistant to degeneration due to their Ca^2+^ buffering capacity [70]. 

In the rat somatosensory cortex, PA is contained only in gamma-aminobutyric acid (GABA) neurons (in hippocampus, cerebellum, and neocortex) [71] (reviewed in [4]). GABA is one of the major inhibitory neurotransmitters in the central nervous system. PA-containing GABAergic interneurons regulate input–output functions in some brain regions. PA seems to take part in regulation of calcium-dependent metabolic and electric processes in such neurons. PA level in soma and neurites of the GABAergic neurons of the brain is high and reaches 50 μM in interneurons [72]. In the neurons located in the brain regions of zebra finch responsible for songs [73] and in neurons located in the optic cortex of cat [74], PA was found in amorphous material, dendrites, and axons; in most nuclei and in association with microtubules; postsynaptic densities; and intracellular membranes.

Caillard et al. [75] suggested that PA may change intracellular Ca^2+^ transients after an action potential. To test this hypothesis, they have applied paired-pulse stimulations (with 30- to 300-ms intervals) at GABAergic synapses between interneurons and Purkinje cells, both in wild-type (PA^+/+^) mice and in PA knockout (PA^−/−^) mice. They found paired-pulse depression in PA^+/+^ mice, but paired-pulse facilitation in PA^−/−^ mice. 1 mM Ca^2+^ buffer EGTA rescued paired-pulse depression in PA^−/−^ mice. The authors concluded that PA can effectively modulate short-term synaptic plasticity.

It has been found that genetic elimination of PA results in changes in GABAergic synaptic transmission, such as an enhancement of synaptic facilitation, a reduction in asynchronous transmitter release, and an increase in the power of gamma oscillations [75,76,77]. PA is thought to act as a slow Ca^2+^ buffer but affects such fast processes as transmitter release at hippocampal and cerebellar GABAergic synapses. Eggerman and Jonas tried to resolve this paradox [78]. They measured PA concentration and paired recordings in rodent hippocampus and cerebellum and found that PA only affects synaptic dynamics in high concentrations approaching its concentration in fast skeletal muscle (~1 mM). According to theoretical evaluations of the authors, although the fraction of free PA (apo-PA) present under physiological conditions is less than 10%, the absolute concentration of the free PA is sufficient to influence fast processes. Note that it is not clear from the text of the article which PA calcium binding constants were used by the authors for these estimates.

Both in neurons and muscle fibers, PA expression strongly depends on their activity. Decreased neuronal activity or oxidative stress in PA containing neurons causes a decrease in PA-immunoreactive neurons, PA messenger RNA, and PA levels [79,80,81].

The effects of PA and mitochondria on the shape of Ca^2+^ transients in neurons are quite similar (reviewed in [69]). It was found that PA and mitochondria demonstrate similarities in their kinetics of Ca^2+^ binding and Ca^2+^ uptake, respectively. Moreover, in all model systems they are regulated in an antagonistic way: an increase in one results in a decrease in the other. This was demonstrated in the case of fast-twitch muscles [82] and Purkinje cells [83] of PA^−/−^ mice, as well as in several cell models, for example, in myotubes [84] and MDCK kidney cells [85].

PA containing neurons are involved in some neuropsychiatric diseases, in particular in schizophrenia and bipolar disorder and in neurodevelopmental disorders including autism spectrum disorder (ASD) [86]. The degradation of PA neurons in ASD is thought to be associated with PA downregulation (reviewed in [69]). This effect was explained by a homeostatic compensation to increase synaptic output of PA neurons [87].

Oxidative stress in PA-GABAergic interneurons and their dysfunction associated with a decrease in the levels of both PA and reduced glutathione (GSH) are hallmarks of the development of schizophrenia [81,88,89,90,91]. Since total antioxidant capacity (AOC) of PA may reach the AOC level of GSH, we suggested that PA might modulate intracellular redox equilibria in a calcium-dependent manner [67]. The loss of PA and GSH results in a decrease in the total cellular AOC, which results in an increase in sensitivity of the cells to oxidative stress. The elevation of cytosolic free Ca^2+^ levels induced by oxidation will result in loading of PA with Ca^2+^ ions, which, in its turn, will decrease its AOC [67]. For these reasons, the increase in ROS level should cause an amplification of the oxidative stress in PA-GABAergic interneurons, which causes their oxidation induced damage and finally their loss. Moreover, the oxidation-induced elevation in cytosolic free Ca^2+^ may initiate specific signaling cascades leading to cell death [92].

Mammalian oncomodulin (OM) is the β isoform of PA, which has at least 53% sequence identity with α-PA [93,94]. Initially, OM was found in cancerous tissue as an oncoprotein [95]. Later, OM was revealed in sensory cells of the guinea pig cochlea [96,97,98]. OM is expressed mostly in cochlear outer hair cells and in vestibular hair cells [99,100,101]. OM was also found in immune cells [102,103,104].

### 5.3. Oncomodulin

Essential features of mammalian OM are similar to those of most other vertebrate β-PAs, such as a length of 109 amino acid residues, isolectric point < 5.0, and a cysteine at position 19. In spite of this, it has been found that mammalian OM and lower vertebrate β-PAs belong to different phylogenetic lineages (reviewed in [96]). In vertebrates, millimolar concentration of OM in outer hair cells is comparable with the millimolar concentration of PA in fast-twitch muscle fibers [105], which means that OM is an effective Ca^2+^ buffer in these cells.

Solution structures of Ca^2+^-free and Ca^2+^-bound rat OM [26] show that removal of Ca^2+^ from OM results in structural alterations (substantial reorganization of the C, D, and E helices), which are more pronounced in comparison with those caused by Ca^2+^ removal from α-PAs. This is illustrated by Figure 4 comparing the 3D structures of Ca^2+^-bound and Ca^2+^-free forms of rat OM (Figure 4A,a, respectively) with the holo- and apo-forms of rat α-PA (Figure 4B,b), as well as Ca^2+^-bound forms of pike α-PA (Figure 4C,c) and pike β-PA (Figure 4D). Comparison of all of these 3D structures show remarkable structural similarity of these proteins. Therefore, although the presence of noticeable Ca^2+^-driven reorganization of OM is discussed, the actual structural changes are not easily detectable by the naked eye. This is further illustrated by Figure 4E representing aligned structures of all of these proteins.

While α-PAs have two Ca^2+^ binding sites with approximately equal affinities (reviewed in [1,2,3]), the Ca^2+^ binding sites of OM are characterized by essentially different affinities [106] (reviewed in [94]). Thus, unlike PA, which has two Ca^2+^/Mg^2+^ sites, OM has one Ca^2+^/Mg^2+^ site and one that is more Ca^2+^ specific. Climer et al. [94] suggested that OM is not pure Ca^2+^ buffer, but it may act as a Ca^2+^ sensor under the right physiological conditions.

Analysis of recent experimental data showed that OM may play an ambiguous role that depends upon the cell type in which it is expressed [94]. Experiments on deletion of OM showed that it is not essential to cochlear development [101]. At the same time, it was found that OM knockout mice start to lose their hearing at 1–2 months and are essentially deaf after 3–4 months. Since progressive hearing loss at 2 months occurs prior to the loss of hair cells in the OM knockout mice, the authors suggested that OM could protect against hearing loss [101]. Triple knock out of three Ca^2+^ buffers, PA, calbindin 1, and calbindin 2, resulted in only minor impacts on hearing [107], which shows that these Ca^2+^ buffers cannot replace OM function. Climery et al. [94] suggested that, in sensory cells, OM maintains auditory function, most likely affecting outer hair cells’ motility mechanisms. They proposed that OM takes part in regulation of outer hair cells elongation (contraction) and shortening (relaxation) mechanisms associated with the cortical lattice, which determines outer hair cells stiffness and electromotility.

OM has been found in macrophages and neutrophils. Benowitz et al. have studied the role of OM in an inflammatory-mediated nerve regenerative model [102,108,109,110]. Analysis of these data shows that, in immune cells, OM seems to be secreted in response to inflammation and its function in this case is facilitation of axon regeneration [94]. Secreted OM either binds to a surface receptor or enters neurons using endocytosis, but only when intracellular cAMP is above basal levels.

Initially, OM was found in several types of mouse, rat, and human tumors, which made it attractive as a potential cancer marker [95]. Later on, it became clear that some normal tissues also contain OM and not all human tumor cell lines contain OM [111]. That is why interest in using this protein as a potential cancer has disappeared.

OM does not cause transformation of cultured cells [112], but neoplastic transformation of rodent cell lines elevates the expression of OM [95,113,114,115,116]. OM expression may be suppressed in non-transformed cells, but can be increased in transformed cells. In order to reveal OM function in tumors, some researchers investigated possible interaction of OM with glutathione reductase, cyclic nucleotide phosphodiesterase, and cell-cycle regulation. Palmer et al. found that purified OM inhibited glutathione reductase in the presence of Ca^2+^ [117]. Glutathione reductase is an enzyme responsible for maintaining reducing conditions within cells. The results of this early study are interesting in connection with a study by Permyakov et al. [67] that showed that PAs have strong antioxidant properties. Klee and Heppel [118] and Clayshulte et al. [119] did not find any modulatory effect of OM on cyclic nucleotide phosphodiesterases.

Two proteins were isolated from frog cutaneous mucus proteome, which take part in prey recognition by snakes of the genus Thamnophis [120]. These proteins, members of the PA family, act as Ca^2+^/Mg^2+^ dependent chemoattractants. They elicit the vomeronasal organ-mediated predatory behavior in *Thamnophis marcianus*. These results show that PAs’ locations are not strictly intracellular, but that they can also be found in exocrine secretions.

High PA immunoreactivity was found in the Leydig cells during testicular development at the stages of intensive testosterone synthesis [121]. PA levels become low at the stages of low Leydig cell activity. It suggests that PA may be involved in the production of testosterone in the Leydig cells. Furthermore, PA immunoreactivity was found in the seminiferous tubules in maturing spermatids and in several other endocrine glands—such as ovaries, pituitary, thyroid, parathyroid, and adrenal glands—which suggests that PA might play some role in endocrine secretions [122,123,124].

It has been found that PA is selectively expressed in the early distal convoluted tubule (DCT) of kidney [125]. PA is colocalized with the thiazide-sensitive Na^+^-Cl^-^ cotransporter (NCC) in the early DCT. Genetically modified (PA^−/−^) mice showed increased diuresis and kaliuresis at baseline with higher aldosterone levels and lower lithium clearance. Endogenous NCC expression was found to be Ca^2+^-dependent and modulated by PA expression level. It has been concluded that, in this system, PA regulates the expression of NCC by modulating intracellular Ca^2+^ signaling in response to ATP in DCT cells [125,126].

## 6. Intrinsic Disorder, Structural Flexibility, and Multifunctionality of Parvalbumin

Although no target proteins have been described for the rat PA, human oncomodulin was reported to interact with mitochondrial creatine kinases 1A and 1B, syndecan binding protein (syntenin), ubiquitin-conjugating enzyme E2A, and WAP four-disulfide core domain 1 (https://thebiogrid.org/576309; accessed on 23 April 2022). Furthermore, analysis human oncomodulin by STRING platform [127] that utilizes seven types of evidence to build the protein–protein interaction (PPI) network for a query protein—such as neighborhood evidence, fusion evidence, experimental evidence, co-occurrence evidence, database evidence, text mining evidence, and co-expression evidence—revealed that this protein can interact with at least 32 partners.

Figure 5 shows that human PA is located at the center of tightly connected PPI network, members of which are linked via 137 interactions. Therefore, on average, each representative of this network interacts with 8.3 partners and together they form a highly interactive system characterized by the average local clustering coefficient of 0.713. Note that the local clustering coefficient of 1 characterizes a network, where every neighbor connected to a given node/protein is also connected to every other node/protein within the neighborhood.

Since multifunctionality and binding promiscuity are considered as specific features of intrinsically disordered proteins of hybrid proteins contacting disordered and ordered regions/domains, we looked at the disorder propensity of human oncomodulin (as an illustrative member of the oncomodulin family) using a set of commonly utilized disorder predictors—such as PONDR^®^ VLXT [128], PONDR^®^ VLS2 [129], PONDR^®^ VL3 [130], PONDR^®^ FIT [131], IUPred2 (Short), and IUPred2 (Long) [132,133]. Results of this analysis are summarized in Figure 6A, which shows that this protein contains significant levels of intrinsic disorder. Most noticeable is highly disordered status of 12 N-terminal and 35 C-terminal residues. Furthermore, as per the outputs of PONDR^®^ VLS2B [129], which was recognized as predictor no. 3 among 43 disorder predicting tools participated in the recently conducted ‘Critical assessment of protein intrinsic disorder prediction’ (CAID) experiment [134], 73.4% residues in human oncomodulin are predicted as disordered, and the entire protein is characterized by the average disorder score (ADS) of 0.59 ± 0.12, which place oncomodulin in the category of highly disordered proteins.

This classification is based on the accepted practice of grouping proteins based on their levels of ADS and/or percent of predicted disordered residues (PPDR). Here, proteins are considered as highly ordered, moderately disordered, and highly disordered if they satisfy the following criteria: ADS < 0.15 (PPDR < 10%), 0.15 ≤ ADS < 0.5 (10% ≤ PPDR < 30%), and ADS ≥ 0.5 (PPDR ≥ 30%), respectively [135]. One might ask an important question as to how come a highly disordered protein such as a PA, putatively quite flexible, has been crystallized and shows a not so high B-factor? The answer to this question can be found in the remarkable dependence of the PA structures on the environmental conditions in general, and on the interaction of these proteins with metal ions in particular. This is well illustrated by the aforementioned conformational behavior of pike PA, which—being intrinsically disordered in its apo-form (similarly to the human OM)—folds in the presence of metal ions into a 3D structure, the stability of which is critically dependent on the nature of the bound metal ions. As far as human OM is concerned, no X-ray crystal structure is available for this protein, and solution NMR structure was solved for its Ca^2+^-bound form in the presence of 100 mM NaCl.

Figure 6A also shows that, according to the outputs of all predictors (with the exception to PONDR^®^ VLXT), all residues in oncomodulin are characterized by the disorder scores exceeding 0.15, suggesting that the entire protein is expected to be either flexible or disordered. This important observation suggests that the unique 3D structures reported for PAs are stabilized by interaction of these proteins with metal ions.

To shed more light on the potential role of intrinsic disorder in functionality of PAs, we used D^2^P^2^ platform (http://d2p2.pro/, accessed on 23 April 2022) [136], which in addition to the information on the disorder status of a query protein generates outputs showing localization of sites of various posttranslational modifications (PTMs) and positions of the intrinsic disorder-based binding sites, molecular recognition features (MoRFs), which are disordered protein regions that undergo disorder-to-order transition at interaction with specific binding partners. Figure 6B represents the corresponding functional disorder profile and shows that human oncomodulin contains one MoRF (residues 64–70) and four phosphorylation sites (residues Ser73, Thr79, Ser81, and Thr83). Furthermore, PAs can be acetylated at N-terminal serine residue. Therefore, PAs can not only bind peptides as discussed in the previous sections, but they also can use their intrinsic disorder for interaction with target proteins. Further analysis is required for better understanding of the roles of intrinsic disorder and structural flexibility in function of these interesting proteins.

Taken together, data presented in this review indicate that PAs can serve as illustrative examples of the “protein structure-function continuum” model [144,145,146,147,148,149] based on the proteoform concept, which suggests that a single gene can encode for multiple structurally and functionally distinct protein molecules—proteoforms, originating as a result of alternative splicing and PTMs [150]—due to the presence of intrinsic disorder and related capability to undergo functional disorder-to-order transitions, as well as because of the structural perturbations induced by functioning [144]. In fact, it is clear that during their functional life, PAs have at least two global possibilities to undergo disorder-to-order transitions, first at binding of metal ions leading to the stabilization of the EF-hands and overall protein fold, and second, at interaction with the partner proteins. Functionality of PAs is further enhanced and controlled by PTMs.

## 7. Conclusions

Despite several decades of study of PA, its physiological functions are still poorly known. This is surprising since the concentration of PA in some organs and tissues reaches a millimolar level. No target proteins have been revealed for PA so far. It is believed that PA acts as a slow calcium buffer. PA concentration in muscles correlates with the speed of their contraction and relaxation. Numerous experiments on various muscle systems have shown that PA accelerates the relaxation of fast skeletal muscles, not during a single contraction–relaxation cycle, but after tetanus. The effect is not very pronounced and PA gene knock out results in about 30% lower rate constant of Ca^2+^ decay. Since fast muscles with high PA concentrations are characterized by highly efficient oxygen uptake, it was suggested that one more physiological function of PA in fast muscles could be a protection of these cells from reactive oxygen species. It was found that the oxidation of the PA by reactive oxygen species is conformation-dependent: antioxidant capacity (AOC) value for apo-PA 4–11-fold exceeds that for the Ca^2+^-loaded protein, while Mg^2+^-bound PA has AOC similar to that of apo-PA.

The role of PA in neurons is even less understood. In the rat somatosensory cortex, PA is contained only in gamma-aminobutyric acid (GABA) neurons (in hippocampus, cerebellum, and neocortex). PA is thought to regulate calcium-dependent metabolic and electric processes within such neurons. Genetic elimination of PA results in changes in GABAergic synaptic transmission, including an enhancement of synaptic facilitation, a reduction in asynchronous transmitter release, and an increase in the power of gamma oscillations.

Mammalian oncomodulin (OM), the β isoform of PA, is expressed mostly in cochlear outer hair cells and in vestibular hair cells. It has been found that OM knockout mice start to lose their hearing at 1–2 months and are essentially deaf after 3–4 months. For this reason, it was suggested that, in sensory cells, OM maintains auditory function, most likely affecting outer hair cells motility mechanisms. In immune cells, OM seems to be secreted in response to inflammation and its function in this case is facilitation of axon regeneration.

All of the above allows us to conclude that the elucidation of the physiological role of parvalbumins in various tissues and organs is undoubtedly worth continuing.

## Data Availability

Not applicable.

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
