# Peer review of "What Is Parvalbumin for?"

_biomolecules, 2022, doi:10.3390/biom12050656_

Round 1

Reviewer 1 Report

In this manuscript, the authors present a comprehensive review on parvalbumin, highlighting structural and functional features and reviewing studies that suggest specific biological roles for the protein that are currently unknown.

The manuscript is generally well-written, but some tables and other forms of graphical summary would aid understanding of the text, particularly when it goes into details that are difficult to focus on without looking at the original data.

Here are some points that should be addressed before the manuscript can be considered ready for publication:

- line49 and following: here and elsewhere in the manuscript, the authors refer to various members of the PA family. From a first reading, it is not clear that the reported variance (106-113 amino acids in length and 11-12 kDa) is actually due to different isoforms/species. A table showing the main characteristics of each PA in the various species (a.a. length, isoelectric points, alpha or beta sublineage,  known or putative interactors  and references) would help understanding. I would recommend adding such a table. This would be helpful in many parts of the text, e.g when explaining details on mammalian OM (lines 467 and following)

- lines 84 and following: it would be helpful to report a multiple sequence alignment of the cited PA forms, to assess the conservation of R85 and E81 in the cation binding loop, as well as Cys residues and black and grey clusters. Such MSA could also be limited to these structural regions.

 - line 92: since Cluster I is apparently conserved within the EF-hand family, it would be good to specify the origin of the CH-pi and CH-O hydrogen bonds. Do they come from conserved residues side chains or backbone? If from the side chain, they could be indicated in the MSA described in the point above. 

- line 138-139. Please indicate the source of such disagreement and the evidence of other mechanisms than sequential binding (cooperative binding? please report references)

- Line 101 and following: Whats is PONDR VSL2P? I guess it stands for "Predictor of intrinsic disordered regions". Not every reader is familiar with this or similar tools. Please explain in a few words how the tool works and how reliable it is. 

- Lines 224-230.  What's the difference in sequence identity among these PA's? Can you comment on what could be the reason for such a discrepancy?

- Line 240. Again, pleas define (earlier) PONDR

- Line 289: this sentence sounds odd. Maybe: Of the possible biological functions of PA, perhaps the one that has been explored the most is that in muscle cells.

- Line 390: please define these acronyms: ORAC, TEAC

- Lines 479 and following. A figure showing the three-dimensional structures of Ca2+-free and Ca2+-bound rat OM compared to the same forms of other PA's would be insightful.

- Lines 568 and following: the analysis of intrinsically disordered regions in PA is very interesting, and the authors did a great job in presenting this aspect. However, considering the many available X-ray structures of parvalbumin (as judged by PDB entries, most of which cover a majority of the protein primary sequence) the reader could find it strange to read that "73.4 % residues in human oncomodulin are predicted to be disordered.... which place oncomodulin to the category of highly disordered proteins". How come that a highly disordered protein, putatively quite flexible, has been crystallized and shows not so high B-factor? The authors could spend a few words on this.

- 7 Conclusions: Rather than reporting a shorter version of the whole manuscript, the conclusion paragraph should focus on just a few points that deserve further studies. I would recommend to shorten this part and make it more focused. 

Author Response

Reviewer #1

Here are some points that should be addressed before the manuscript can be considered ready for publication:

- line49 and following: here and elsewhere in the manuscript, the authors refer to various members of the PA family. From a first reading, it is not clear that the reported variance (106-113 amino acids in length and 11-12 kDa) is actually due to different isoforms/species. A table showing the main characteristics of each PA in the various species (a.a. length, isoelectric points, alpha or beta sublineage,  known or putative interactors  and references) would help understanding. I would recommend adding such a table. This would be helpful in many parts of the text, e.g when explaining details on mammalian OM (lines 467 and following)

RESPONSE: Thank you for this very useful suggestion. The corresponding table is added to the revised manuscript.

- lines 84 and following: it would be helpful to report a multiple sequence alignment of the cited PA forms, to assess the conservation of R85 and E81 in the cation binding loop, as well as Cys residues and black and grey clusters. Such MSA could also be limited to these structural regions.

RESPONSE: Thank you for pointing this out. The corresponding information is added to the revised manuscript as a new Figure 2.

 - line 92: since Cluster I is apparently conserved within the EF-hand family, it would be good to specify the origin of the CH-pi and CH-O hydrogen bonds. Do they come from conserved residues side chains or backbone? If from the side chain, they could be indicated in the MSA described in the point above. 

RESPONSE: Thank you for pointing this out. The corresponding information is shown in a new Figure 2.

- line 138-139. Please indicate the source of such disagreement and the evidence of other mechanisms than sequential binding (cooperative binding? please report references)

RESPONSE: Thank you for pointing this out. The corresponding information is added to the revised manuscript (see lines 222-225)

- Line 101 and following: What is PONDR VSL2P? I guess it stands for "Predictor of intrinsic disordered regions". Not every reader is familiar with this or similar tools. Please explain in a few words how the tool works and how reliable it is. 

RESPONSE: Thank you for pointing this out. The corresponding clarification is added.

- Lines 224-230.  What's the difference in sequence identity among these PA's? Can you comment on what could be the reason for such a discrepancy?

RESPONSE: Information on sequence identity of five PAs is presented in new Figure 2. In our view, there is no obvious sequence-based explanation for the observed differences among these PAs.

- Line 240. Again, pleas define (earlier) PONDR

RESPONSE: Definition of PONDR was provided.

- Line 289: this sentence sounds odd. Maybe: Of the possible biological functions of PA, perhaps the one that has been explored the most is that in muscle cells.

RESPONSE: Thank you for this suggestion. The corresponding sentence was corrected as recommended.

- Line 390: please define these acronyms: ORAC, TEAC

RESPONSE: Definitions of these acronyms are added to the revised manuscript.

- Lines 479 and following. A figure showing the three-dimensional structures of Ca2+-free and Ca2+-bound rat OM compared to the same forms of other PA's would be insightful.

RESPONSE: Following your recommendations, we added a new Figure 4 containing structures for holo- and apo-forms of OM and other PAs. Comparison of these 3D structures show remarkable structural similarity of all these proteins. Therefore, although the presence of noticeable Ca2+-driven reorganization of OM is discussed, the actual structural changes are not easily detectable by the naked eye. This is further illustrated by Figure 4E representing aligned structures of all these proteins.

- Lines 568 and following: the analysis of intrinsically disordered regions in PA is very interesting, and the authors did a great job in presenting this aspect. However, considering the many available X-ray structures of parvalbumin (as judged by PDB entries, most of which cover a majority of the protein primary sequence) the reader could find it strange to read that "73.4 % residues in human oncomodulin are predicted to be disordered.... which place oncomodulin to the category of highly disordered proteins". How come that a highly disordered protein, putatively quite flexible, has been crystallized and shows not so high B-factor? The authors could spend a few words on this.

RESPONSE: Thank you for this important question. The answer to it can be found in the remarkable dependence of the structures of PAs on the environmental conditions in general, and on the interaction of these proteins with metal ions in particular. This is well illustrated by the aforementioned conformational behavior of pike PA, which, being intrinsically disordered in its apo-form (similarly to the human OM), folds in the presence of metal ions into a 3D structure, stability of which is critically dependent on the nature of the bound metal ions. As far as human OM is concerned, no X-ray crystal structure is available for this protein, and solution NMR structure was solved for its Ca2+-bound form in the presence of 100 mM NaCl. This discussion is added to the revised manuscript.

- 7 Conclusions: Rather than reporting a shorter version of the whole manuscript, the conclusion paragraph should focus on just a few points that deserve further studies. I would recommend to shorten this part and make it more focused.

RESPONSE: Thank you for pointing this out. The conclusion section was rewritten as recommended.

Reviewer 2 Report

The manuscript entitled “What Is Parvalbumin for?” deals with structural and physicochemical properties and potential physiological functions of the two isoforms of parvalbumin: alpha-parvalbumin and beta-parvalbumin/oncomodulin. The authors also have discussed the protein-protein interaction network and the putative disordered regions of beta-parvalbumin/oncomodulin. The manuscript provides comprehensive and relevant information on parvalbumins, playing various possible functions in vertebrates. I have only minor specific comments:

In Figures 1 and 2, it would be nice to include sequence information for the two isoforms of rat parvalbumins, showing the position and conservation of the amino acids mentioned in the text, such as calcium ion-coordinating positions, cluster I and cluster II. As Figures 1 and 2 are based on the same PDB data at different angles, it is better to match the color scheme of Figure 2 to that of Figure 1.

Chapter 5 (lines 288-543) is long and not reader-friendly; It would be easier to read if chapter 5 were split into several parts (5-1., 5-2.,…), each with an appropriate title.

Line 390: It should be defined what “ORAC” and “TEAC” are abbreviations for.

In Figure 4, the "B" is in the bottom right-hand corner. Please position it appropriately.

There are several typos: Line 237 “r” is missing “Moreove”; Line 287 “microtubuli” instead of “microtubule”?; Lines 302 and 305 “2+” should be superscripted; Line 611 “Pas” should be “PAs”?

Author Response

Reviewer # 2

In Figures 1 and 2, it would be nice to include sequence information for the two isoforms of rat parvalbumins, showing the position and conservation of the amino acids mentioned in the text, such as calcium ion-coordinating positions, cluster I and cluster II.

RESPONSE: Thank you for pointing this out. To address this issue, we added a new Figure 2, where aligned sequence of PAs are presented.

As Figures 1 and 2 are based on the same PDB data at different angles, it is better to match the color scheme of Figure 2 to that of Figure 1.

RESPONSE: In our view, changing the color schema of Figure 2 will generate confusion, as it will be too colorful.

Chapter 5 (lines 288-543) is long and not reader-friendly; It would be easier to read if chapter 5 were split into several parts (5-1., 5-2.,…), each with an appropriate title.

RESPONSE: Thank you for pointing this out. Following your recommendations, we split section 5 on several parts.

Line 390: It should be defined what “ORAC” and “TEAC” are abbreviations for.

RESPONSE: Definitions of these acronyms are added to the revised manuscript.

In Figure 4, the "B" is in the bottom right-hand corner. Please position it appropriately.

RESPONSE: In our view, “B” is positioned appropriately

There are several typos: Line 237 “r” is missing “Moreove”; Line 287 “microtubuli” instead of “microtubule”?; Lines 302 and 305 “2+” should be superscripted; Line 611 “Pas” should be “PAs”?

RESPONSE: Thank you for pointing this out. The corresponding corrections were introduced to the revised manuscript.

Round 2

Reviewer 1 Report

The authors have addressed all the points raised in my previous review. In my opinion, the manuscript has improved in terms of clarity and is now ready for publication.

This manuscript is a resubmission of an earlier submission. The following is a list of the peer review reports and author responses from that submission.